# OS-Catalyst: Advancing Computer-Using Agents Efficiency through Adaptive Action Compression

## Abstract

Driven by advances in Vision-Language Models (VLMs), computer-using agents have recently demonstrated remarkable capabilities in complex reasoning, software control, and the automation of digital workflows. However, the existing step-by-step paradigm requires extensive interaction with the model, and the resulting query latency emerges as a key bottleneck for real-world adoption. To address this limitation, we propose that agents should be able to output a sequence of actions after each observation, enabling efficient execution without constant model queries. In this work, we introduce OS-Catalyst, a method that transforms standard computer-using models into agents with the capability of action sequence prediction. To enable this, we design a data collection pipeline tailored for compressed action trajectories in computer-using environments. Building on this pipeline, we construct a large-scale dataset within the WorkArena benchmark and train computer-using agents for action sequence prediction. Through extensive experiments, we show that OS-Catalyst enables up to 50% faster task completion on office-related benchmarks without sacrificing success rate.

## 1 Introduction

In recent years, the rapid development of Large Language Models (LLMs) (Anthropic, 2025; OpenAI, 2025a; Bai et al., 2025) has driven the expansion of artificial intelligence from natural language processing (NLP) to a broader range of application domains. Along with this trend, LLM-based agents have gradually become a key point of research in both academia and industry (OpenAI, 2025b; Manus, 2025). These agents not only demonstrate strong capabilities in information processing and knowledge reasoning but also exhibit growing potential for handling complex tasks and executing high-level decision-making through direct interactions with operating systems and applications. In particular, "computer-using agents" (Cheng et al., 2024; Anthropic, 2025; Qin et al., 2025b; Sun et al., 2024b) are designed to simulate human–computer interactions. In practice, this often manifests as agents that operate directly on Graphical User Interfaces (GUI), which perceive dynamic layouts, ground references to interface elements, and plan following actions. In the future, such agents are expected to lower the operational barriers of both daily affairs and professional tasks, thereby advancing the evolution of human–computer collaboration.

Currently, GUI agents (OpenAI, 2025a; Liu et al., 2025c) primarily interact through the step-by-step paradigm shown in figure 1 (a). Given a user's instruction, the model processes inputs such as screenshots or accessibility trees, and iteratively produces reasoning and corresponding actions until the task is complete. Multi-agent frameworks (Agashe et al., 2025b; Jia et al., 2024; Wu et al., 2024) follow a similar reasoning paradigm, though additional stages may be introduced during action planning and execution. Such an interaction paradigm requires agents to execute 10–30 steps to complete a single GUI task. Querying the model repeatedly accounts for most of the time in GUI tasks. Consequently, completing a single GUI task often takes at least several minutes, which introduces nontrivial bottlenecks for real-world deployment and commercial adoption.

However, we observe that in many scenarios, especially office-related tasks, this interactive paradigm leaves significant room for compression. In fact, it is often unnecessary to request a new observation before every single action. For humans, when completing tasks such as filling out a form, a single

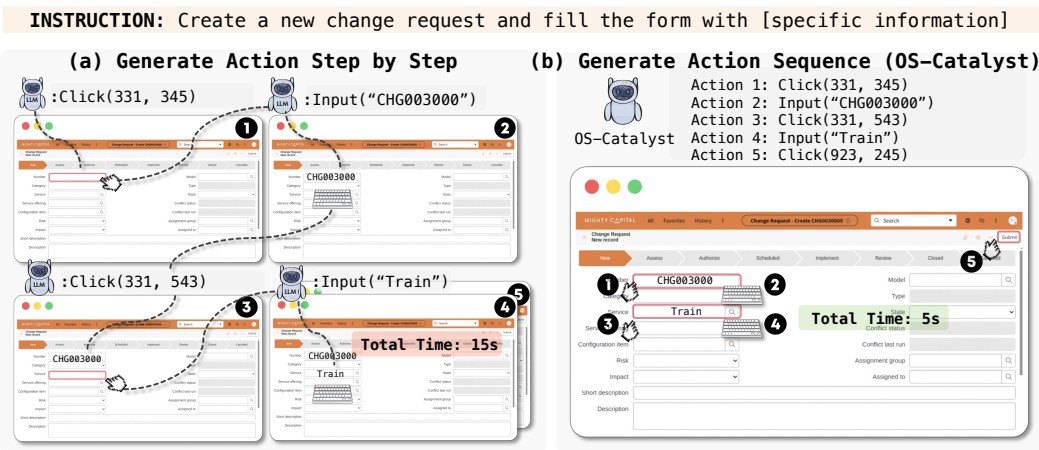

Figure 1: Main idea of OS-CATALYST. Traditional GUI agents generate actions step by step (a), which incurs repeated model–environment interactions and high latency. OS-CATALYST (b) enables the model to predict multiple valid actions in one step, thus compressing trajectories and improving execution efficiency.

round of observation is often sufficient to determine the type and location of several subsequent actions as shown in figure 1 (b). Inspired by this observation, we propose OS-CATALYST, which introduces a novel interaction paradigm for GUI agents. OS-CATALYST integrates action-sequence prediction idea, a tailored data compression pipeline, and fine-tuning strategies, enabling models to acquire the ability to output coherent multi-step action sequences from a single observation.

We conduct measurements on WorkArena (Drouin et al., 2024), a benchmark designed for GUI-based office tasks. The result reveals that if the model outputs the maximal feasible sequence of actions after each observation, at least 50% of task execution steps can be eliminated. However, current models are unable to correctly infer such multi-step action sequences through prompt-based guidance alone. This indicates that existing GUI agents lack the ability to reason about longer action plans within a given observation. To address this, we construct a dataset within the WorkArena environment and train models based on UI-TARS (Qin et al., 2025b). OS-CATALYST achieved up to a 50% reduction in task execution times compared with step-by-step paradigm.

- We propose a novel direction for improving the efficiency of GUI agents through **adaptive action compression**, which reduces unnecessary observations between sequential actions.
- We construct a dataset and develop corresponding models within the WorkArena environment, enabling multi-step action prediction from a single observation.
- We achieve up to **50% reduction** in task execution time on office benchmarks, demonstrating the effectiveness of OS-CATALYST.

## 2 RELATED WORK

**Computer-using agents.** Unlike early LLM-based agents that parse GUIs (Graphical User Interface) into structured text (Deng et al., 2023; Zhou et al., 2024) and navigate through provided tools like programs (Sun et al., 2024a) or API calls (Wu et al., 2024; Zhang et al., 2024a), VLM-based GUI agents directly perceive raw screenshots and output human-like atomic keyboard/mouse operations—markedly boosting adaptability while introducing new challenges. First, VLMs are required to perceive detailed information and localizing elements in high-resolution screenshots. Beyond supervised pre-training on large-scale grounding datasets (Cheng et al., 2024; Chen et al., 2024b; Xu et al., 2024; Gou et al., 2024; Wu et al., 2025c), efforts include training high-resolution processing (Hong et al., 2024; Yang et al., 2024; Li et al., 2024) or token selection (Ge et al., 2024; Wu et al., 2025b; Zhang et al., 2024b) modules for visual encoders, and designing specific reasoning strategies for dynamic focusing or test-time scaling (Wu et al., 2025a; Yang et al., 2025; Liu et al., 2025b). Furthermore, computer-using agents necessitate strong multi-turn planning capabilities (Xie

et al., 2024b; Sun et al., 2025). Two mainstream approaches exist: one involves elaborately designed agentic workflow frameworks for proprietary VLMs (Zhang et al., 2025b; Jiang et al., 2025; Zheng et al., 2024a; Wang et al., 2024; Jia et al., 2024; Agashe et al., 2025a), which comprise multiple external modules such as hierarchical planning, systematic memory organization, and multi-agent collaboration; the other focuses on conducting supervised fine-tuning and reinforcement learning to endow open-source VLMs with native long-horizon reasoning and error recovery capabilities (Wang et al., 2025a; Xia & Luo, 2025; Liu et al., 2025a; Qi et al., 2024).

**Efficiency of agent workflows.** Agentic workflows rooted in the CoT (Wei et al., 2022) and ReAct-style (Yao et al., 2023) paradigms unlock LLMs' capabilities for complex tasks, while simultaneously significantly increasing tool invocation complexity and context length—ultimately leading to higher costs and degraded performance. To address the issue of reasoning inefficiency, within the multi-agent setup, DAAO (Su et al., 2025) leverages the complementary advantages of heterogeneous models and introduces LLM routing based on query difficulty estimator to implement an adaptive orchestration system. Optima (Chen et al., 2024a) and Puppeteer (Dang et al., 2025) integrate the balance between performance and efficiency into reward functions, continuously enhancing the system's dynamic orchestration and adaptive evolution capabilities through reinforcement learning.

In the specific context of computer-using tasks, OS-Copilot (Wu et al., 2024), Mobile-Agent-E (Wang et al., 2025b), and AppAgentX (Jiang et al., 2025) excavate repetitive patterns from historical actions, organize them into shortcuts or tool scripts, and store these in long-term memory to enable reuse and improve efficiency. Similarly, UFO[2] (Zhang et al., 2025a) incorporates speculative multi-action output; yet the complexity of GUI environments lies in the fact that target elements shift unpredictably with interactions, necessitating system API calls to ensure robust execution. Likewise, Dyna-Think (Yu et al., 2025) demonstrates effective multi-action reasoning refinement under accessibility-tree–based UI representation, leveraging structured textual information of interface elements. However, current models that rely purely on visual observations still struggle to achieve comparable prediction quality, as they lack explicit semantic and hierarchical cues. OSWorld-Human (Abhyankar et al., 2025) recognizes this limitation and provides manually grouped action annotations as a benchmark to assess efficiency, while mainly focusing on evaluation. Building on this insight, we move one step closer by internalizing the prediction of environmental dynamics into the model through large-scale supervised training.

# 3 METHOD

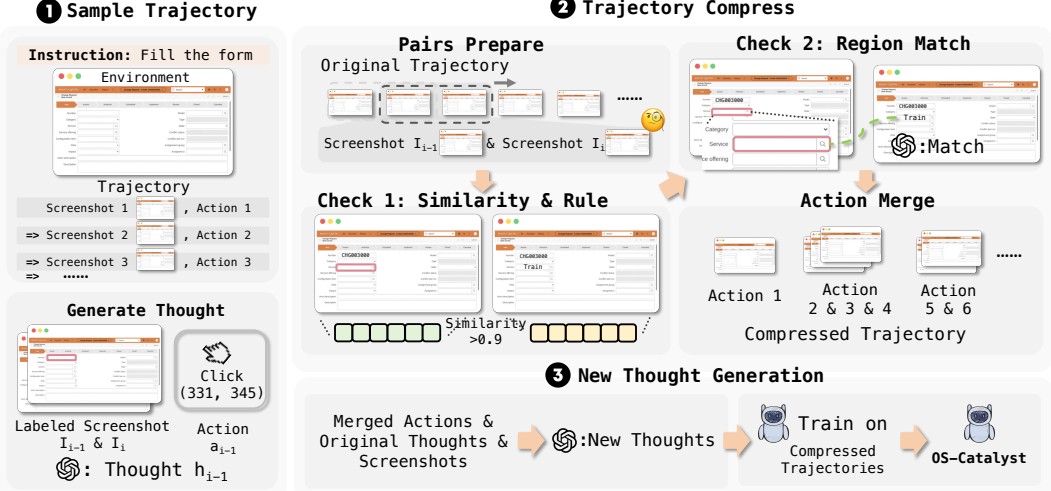

Figure 2: Data construction pipeline of OS-CATALYST. From sampled trajectories, we merge consecutive actions validated by similarity and region checks, regenerate thoughts, and fine-tune the model on the compressed trajectories.

We propose OS-CATALYST, a method that enables GUI agents to improve efficiency by adaptively predicting action sequences. In the following, we detail the components of OS-CATALYST. We first introduce the formulation of action sequences and describe how they are executed in the environment. Next, we present the dataset construction pipeline, which produces both step-level and compressed trajectories. Finally, we present the overall process of training our model.

## 3.1 ACTION SEQUENCE FORMULATION

Most GUI agents today work step by step: the model outputs one action(or a fixed combination of two actions, for example, a click followed by a type), the environment executes it, and then the model is prompted again to predict the next action (Sun et al., 2024b; Gou et al., 2024). This process repeats until the task is finished. While simple, this approach is often inefficient. For example, imagine a form that requires filling multiple fields. A human can look at the page once and immediately know the next several operations, such as *clicking field A $\rightarrow$ typing the name*, then *clicking field B $\rightarrow$ typing the gender*, and so on.

Inspired by this, we expect the model to also predict multiple consecutive actions in advance, rather than only one action at a time. In particular, the model should be able to adaptively decide the length and content of the sequence according to the task requirement, the current interface, and the execution progress. We define an action sequence as a short list of consecutive actions predicted by the model at once. The environment still executes the actions one by one in order, but the model is only prompted again after the whole sequence has been executed. This design reduces the number of model-environment interactions to shorten the overall time needed for task execution.

Let $s_t$ be the environment state at step $t$. The model first generates a thought $h_t$, which describes its plan for the next steps, and then outputs an action sequence

$$A_t = \{a_{t_1}, a_{t_2}, \ldots, a_{t_k}\}, \quad k \leq K,$$

where $a_i$ is an atomic action and $K$ is the maximum sequence length. The environment executes $A_t$ sequentially:

$$s_{t+1} = T(s_t, a_{t_1}, a_{t_2}, \ldots, a_{t_k}),$$

where $T$ denotes the process that interprets the model's action output and applies the corresponding action in the environment. After $A_t$ is finished, the model receives $s_{t+1}$ and predicts the next pair $(h_{t+1}, A_{t+1})$.

## 3.2 DATASET CONSTRUCTION

We attempt to use prompts to let the model output multiple actions at first. However, we found two major problems. First, the model had no awareness of producing action sequences and would not attempt multi-step actions within the current interface. Second, even when action sequences were produced, the accuracy was very low. Therefore, we aim to construct a dataset for post-training, in order to enhance the model's ability in multi-step planning and action prediction. The complete dataset construction process is shown in Figure 2.

**Raw Trajectory Collection.** WorkArena (Drouin et al., 2024) is an enterprise software environment built on the ServiceNow[1] platform, designed to evaluate GUI agents on realistic knowledge-work tasks such as form filling, list filtering and sorting, information retrieval from knowledge bases, service catalog usage, and menu navigation. It provides multimodal observations of the interface (e.g., HTML, accessibility tree, and screenshots). Each task comes with natural language instructions and automatic checks for the final task completion.

In this environment, each type of task is defined by a task template. By randomly sampling the conditions and input values within a template, multiple task instances can be constructed. The authors of WorkArena also provide a `cheat_function`, which generates the correct Playwright action trajectory based on the specific configuration of a task. This function enables us to obtain reliable ground-truth trajectories for training.

We collect a total of 420 trajectories across 21 tasks. Each trajectory contains the sequence of environment states (including screenshots) along with the corresponding atomic actions(click, type,

---

[1]https://www.servicenow.com/

select, etc.) and their associated bounding boxes, forming the raw data for subsequent action sequence compression and training.

Formally, let a raw trajectory be $\tau = \{(s_i, a_i)\}_{i=1}^{T}$, where $s_i$ is the environment state (including screenshot $I_i$ and other structural views) at step $i$, and $a_i$ is the atomic action with its bounding box.

**Thought Generation.** Previous work has shown that explicitly modeling the reasoning process can improve the inference ability of GUI agents (Qin et al., 2025b; Zhang et al., 2025c). Following this idea, we augment raw trajectories with an additional *thought* before each action. The *thought* reflects the agent's consideration of what to do next, serving as a small plan that guides the subsequent actions.

To generate a thought $h_i$ for action $a_i$, we provide GPT-4o (Hurst et al., 2024) with the screenshot pair $(I_i, I_{i+1})$ together with the executed action $a_i$. In the screenshots, the bounding box of the element involved in $a_i$ is highlighted with a red rectangle, which helps the model identify the relevant interface element. The model then produces a natural language description that explains the intention of $a_i$ by referring to how it changes the interface from $s_i$ to $s_{i+1}$. The augmented trajectory is thus $\tilde{\tau} = \{(s_i, h_i, a_i)\}_{i=1}^{T}$.

In this way, each action is aligned with both its execution context and a reasoning thought, providing extra supervision to support task understanding.

**Trajectory Compression Pipeline.** In GUI tasks, action sequences cannot be arbitrarily compressed, since some actions may trigger page transitions or significant interface changes. Such changes make it impossible to accurately anticipate the location of the next action without observing the updated interface. To obtain trajectories with action sequences, we design a compression pipeline that transforms raw step-by-step trajectories into compressed ones, as shown in the right of Figure 2. Let a raw trajectory be $\tau = \{(s_i, h_i, a_i)\}_{i=1}^{T}$ from WorkArena. We sequentially check whether two adjacent steps can be output within the same sequence. The prerequisite is that the result of the first action does not affect the element involved in the second one. If the previous action includes navigating to another page or scrolling to reveal hidden content, the condition is not satisfied.

*Pair preparation.* From $\tau$ we build adjacent candidates consisting of $\mathcal{P} = (s_t, a_t, s_{t+1}, a_{t+1})$.

*Check 1: Similarity & Rule.* For each pair, we compute the Structural Similarity Index (SSIM) (Wang et al., 2004) between consecutive screenshots $I_i$ and $I_{i+1}$. Only pairs with similarity greater than a threshold (0.9 in our experiments) are retained, filtering out major interface transitions. We also add a restriction that if the first action is a `scroll` or any operation that inevitably changes the page layout, the pair is automatically filtered out.

*Check 2: Region match.* For the remaining pairs, we perform local verification. The key criterion for compression is whether the first action changes the position or state of the element involved in the second action. To verify this, we use the bounding box $b_{i+1}$ of the second action and crop the corresponding regions from screenshots $I_i$ and $I_{i+1}$. We then query GPT-4o with these cropped regions and the action descriptions to decide whether the two actions can be safely merged. This step ensures that the element required by the second action remains stable and does not depend on intermediate model feedback.

*Action merge.* We greedily merge consecutive actions as long as both checks pass, forming a compressed sequence $A_t = \{a_{t_1}, a_{t_2}, \ldots, a_{t_k}\}, k \le K$, with $K = 5$ as the maximum sequence length. The merged sequence is then stored in the compressed trajectory $\hat{\tau}$.

*New thought generation.* Since original thoughts $h_i$ are tied to atomic actions, we regenerate a new thought $\hat{h}_t$ for each compressed sequence $A_t$. To do this, we prompt GPT-4o (Hurst et al., 2024) with the start and end screenshots $(I_i, I_{i+k})$, and the original thoughts. Based on this input, GPT-4o produces a concise description that explains the reasoning behind executing $A_t$. The final compressed trajectory is represented as $\hat{\tau} = \{(s_t, \hat{h}_t, A_t)\}_{j=1}^{M}, \quad A_t = \{a_{t_1}, a_{t_2}, \ldots, a_{t_k}\}, |A_t| \le K$, which is then used for training.

Table 1: Statistics of our constructed datasets. We report the number of trajectories, the average number of steps per trajectory, and the average number of actions per step.

| Dataset | #Trajectories | Avg. Steps / Traj. | Avg. Actions / Step |
|---|---|---|---|
| Work-Step | 420 | 19.00 | 1.00 |
| Work-Seq | 420 | 12.58 (-33.8%) | 1.51 (+51.0%) |

## 3.3 DATASET STATISTICS

The original WorkArena benchmark contains 25 task types. To clearly separate the training and test sets, we select 21 task types for training. For each task type, we collect 20 distinct trajectories using different random seeds, resulting in a total of 420 trajectories. Based on these raw trajectories, we construct two datasets that differ in how the actions are represented and organized.

**Work-Step** is built from the raw trajectories by adding a *thought* to each atomic action, as described in Section 3.2. The dataset keeps the original step-by-step format with reasoning information.

**Work-Seq** is built from Work-Step using the compression pipeline described in Section 3.2. In this process, consecutive actions are merged into a short action sequence when the conditions are satisfied.

Table 1 summarizes the statistics of our constructed datasets. Compared to Work-Step , Work-Seq significantly reduces the average step length to 12.6(-33.8%), due to the increase of the average number of actions per step to 1.51(+51.0%). Further details of the dataset can be found in Appendix D.

## 3.4 TRAINING STRATEGY

To train models to generate coherent action sequences, we adopt a supervised objective that couples *thoughts* with *actions*. Unlike single-step prediction, sequence generation requires stronger reasoning: after proposing the first action, the model must decide whether one or more subsequent actions are still determinable from the current observation and remain unaffected by earlier actions. If so, they can be merged together as a coherent sequence; otherwise, the model should stop and wait for a new observation. Therefore, by training thoughts and actions together, we encourage the model to use the thought component as multi-step planning, enabling it to develop the reasoning ability required for predicting coherent action sequences.

**Context-Aware Formulation.** Another design choice is to include recent interaction history. Using only the current screenshot leaves the model unaware of past actions and task progress, while conditioning on the full trajectory can exceed the context window and introduce noise from distant steps. To balance these trade-offs, we use the last $L = 5$ steps as context, which captures short-term dependencies (e.g., opening a menu before selecting an option) while remaining within the model's effective context length. For each training instance, the model is conditioned on the last $L$ steps of history,

$$\mathcal{C}_t = \{(s_{t-L}, h_{t-L}, A_{t-L}), \dots, (s_{t-1}, h_{t-1}, A_{t-1})\},$$

together with the current state $s_t$. The task is to predict both the next thought $h_t$ and the next action sequence $A_t$, thus capturing both reasoning and execution.

**Thought–Action Training.** We model the joint generation of thought $h_t$ and action sequence $A_t$ as

$$p_\theta(h_t, A_t \mid I_t, \mathcal{C}_t) = \prod_{j=1}^{|h_t|} p_\theta(h_{t,j} \mid I_t, \mathcal{C}_t, h_{t,<j}) \prod_{m=1}^{|A_t|} p_\theta(A_{t,m} \mid I_t, \mathcal{C}_t, h_t, A_{t,<m}).$$

Training uses the standard next-token cross-entropy objective:

$$\mathcal{L}(\theta) = -\sum_{t=1}^{T} \left( \sum_{j=1}^{|h_t|} \log p_\theta(h_{t,j} \mid I_t, \mathcal{C}_t, h_{t,<j}) + \sum_{m=1}^{|A_t|} \log p_\theta(A_{t,m} \mid I_t, \mathcal{C}_t, h_t, A_{t,<m}) \right).$$

Table 2: Evaluation results on WorkArena(Seen). The first three metrics are success rate (SR), partial success rate (PSR), number of action per step (#A/S). We use UI-TARS(Work-Step) as the baseline to compute the relative changes of efficiency metrics for UI-TARS(Work-Seq). For SR, PSR, and efficiency metrics, we highlight the best (bold) and second-best (underline) results.

| Model | SR | PSR | #A/S | Step Time (s) | Task Time (s) |
|---|---|---|---|---|---|
| **7B Models** | | | | | |
| UI-TARS-7B-DPO | 0.036 | 0.072 | 1.00 | 11.66 | 580.80 |
| UI-TARS(prompt) | 0.024 | 0.072 | 1.00 | 7.95 | 381.02 |
| UI-TARS(Work-Step) | **0.095** | 0.267 | 1.00 | 12.00 | 291.41 |
| UI-TARS(Work-Seq) | 0.083 | **0.277** | **1.33** (+33.0%) | 9.20 | **147.90** (-49.2%) |
| **72B Models** | | | | | |
| UI-TARS(Work-Step) | **0.079** | 0.210 | 1.00 | 15.19 | 389.41 |
| UI-TARS(Work-Seq) | 0.060 | **0.294** | **1.20** (+20.0%) | 13.02 | **308.50** (-20.8%) |

# 4 EXPERIMENT

## 4.1 EVALUATION BENCHMARK

**Workarena.** We evaluate our method on WorkArena (Drouin et al., 2024), a benchmark ideal for testing action-compression due to its focus on automating real-world, multi-step business tasks. WorkArena simulates the repetitive, structured workflows of daily office work, such as list operations, form fillings, and service catalog tasks (item purchasing), where employees naturally execute predictable action sequences. This characteristic makes it perfectly suited for assessing our method's efficiency in generating multiple actions per turn, as successfully completing its tasks requires the model to plan and compress these logical sequences into a single, cohesive output.

Since the benchmark contains 25 task types and we use 21 types for training, the first test set (**WorkArena(Seen)**) contains tasks from the same 21 types but generated with different random seeds. This results in 84 distinct tasks that occur in the same scenarios as the training set, but differ in their specific requirements. The second test set (**WorkArena(Unseen)**) is built from the remaining 4 task types that are completely excluded from training. This set includes 16 tasks and serves to evaluate the generalization ability of the models to novel task types.

**OSWorld.** OSWorld (Xie et al., 2024a) is a GUI-based benchmark that evaluates computer-use agents across heterogeneous software environments, including office tools, operating systems, web browsers, and developer applications. It is particularly suited for assessing cross-domain generalization under out-of-distribution settings. In our experiments, we use OSWorld to test whether OS-CATALYST transfers effectively beyond the WorkArena setting.

## 4.2 MODEL SETTINGS

**UI-TARS.** We use UI-TARS (Qin et al., 2025a) as our model. UI-TARS is a GUI agent model. It takes screenshots as input and produces human-like interactions (mouse clicks, keyboard typing, etc.). Unlike many previous systems that rely heavily on prompt engineering or wrapped workflows around large models, UI-TARS is an end-to-end model that unifies perception, grounding, reasoning, and action directly.

## 4.3 BASELINE CONSTRUCTION

We consider four groups of models in our experiments.

1. The original `UI-TARS-7B-DPO` and `UI-TARS-72B-DPO` models with their default prompting setup, which predict the next action step by step.

2. The same UI-TARS models prompted to output an action sequence on each page, without any additional training, denoted as **UI-TARS(prompt)**.

3. A fine-tuned version of UI-TARS trained on the Work-Step dataset, which provides step-by-step trajectories augmented with thoughts, denoted as **UI-TARS(Work-Step)**.

4. A fine-tuned version of UI-TARS trained on the Work-Seq dataset, which contains compressed action sequences with thoughts, i.e., **UI-TARS(Work-Seq)**.

## 4.4 Metrics

**Task Success.** We evaluate success rate using the rule-based outcome validator built in WorkArena. However, the task set in WorkArena are long-tailed, often requiring over 30 steps on average and making it overly difficult for current GUI models. To better capture the reasoning capability of models under such challenging settings, we further implement a Partial Success Rate (PSR) validator for each task, as PSR can more fairly reflect partial progress and provide a more informative measure of model performance. For tasks on lists, we average the total 1.0 point on each input box of the filter panel. For tasks on forms, we reserve 0.2 point for submitting the form with all items filled correctly and the rest 0.8 point is averaged on all items that need to be filled. For tasks on service catalogs, since the general procedure is firstly navigating to the web page of the requested item, then filling some requested configurations and finally submitting, we reserve 0.3 point for successful navigation and 0.1 point for successful submission, and the rest 0.6 point is averaged on the requested configurations.

**Efficiency Metrics** We also report three metrics that measure execution efficiency. A/S (actions per step) denotes the average number of valid actions contained in each model output. To ensure fairness, steps where no executable action is produced are excluded from this calculation. Step Time (s) measures the average latency of generating one model output, while Task Time (s) reflects the average time to finish a task, which accumulates both step-level latencies and the number of steps.

## 5 Main Result and Analysis

### 5.1 How Does OS-Catalyst Improve Efficiency?

**WorkArena(Seen) Result.** For both 7B and 72B settings, models trained with our Work-Seq dataset achieve notable improvements in execution efficiency compared to those trained on Work-Step. Specifically, UI-TARS(Work-Seq) reduces the average task time from 291.4s to 147.9s in the 7B case (nearly a 50% reduction), and from 389.4s to 308.5s in the 72B case, while maintaining comparable success rates. Compared to the base model (UI-TARS-7B-DPO) and the prompt-only variant (UI-TARS(prompt)), our method delivers efficiency gains through action sequence compression.

Both baselines almost never output multiple actions in a single step (average actions per step = 1.0), while UI-TARS(Work-Seq) improves its efficiency by performing multiple actions within each step. By increasing the average actions per step to 1.33 (7B) and 1.20 (72B), our model effectively reduces unnecessary model–environment interactions, leading to substantial savings in overall task duration.

**WorkArena(Unseen) Result.** In addition, we report results on the Unseen set containing four task types that were not used for training. On this set, UI-TARS(Work-Seq) also maintains efficiency advantages to other baselines. For 7B models, UI-TARS(Work-Seq) reduces the average task time to 135.1s, compared to 157.4s for UI-TARS(Work-Step) and over 300s for the base model. A similar trend is observed in the 72B models, where task time decreases from 405.9s to 360.3s. UI-TARS(Work-Seq) also achieves higher average actions per step (1.28 for 7B and 1.25 for 72B) in unseen set, indicating that the ability to predict action sequence generalizes beyond the training tasks and leads to consistent efficiency gains on unseen tasks.

**Action Sequence Type.** We further analyze the distribution of action sequence types produced by UI-TARS(Work-Seq). As shown in Figure 3, most of the generated sequences have two actions, though we also observe successful cases with three or even four actions. The most frequent patterns are [click, type] and [click, click], which closely correspond to common GUI interaction routines, such as selecting a field followed by text input or navigating option menus through consecutive clicks. Other action sequence types also carry concrete meaning, such as [click, type, click] for filling in a field followed by confirmation, and [click, type, click,

Table 3: Evaluation results on WorkArena(Unseen). We use UI-TARS(Work-Step) as the baseline to compute the relative changes of efficiency metrics for UI-TARS(Work-Seq). For SR, PSR, and efficiency metrics, we highlight the best (bold) and second-best (underline) results. Step Time values are reported without relative changes.

| Model | SR | PSR | A/S (non-empty) | Step Time (s) | Task Time (s) |
|-------|-----|-----|-----------------|---------------|---------------|
| **7B Models** | | | | | |
| UI-TARS-7B-DPO | **0.063** | 0.125 | 1.00 | 6.35 | 309.80 |
| UI-TARS(prompt) | **0.063** | 0.100 | 1.00 | 10.84 | 500.48 |
| UI-TARS(Work-Step) | 0.000 | **0.157** | 1.00 | 9.12 | 157.38 |
| UI-TARS(Work-Seq) | **0.063** | 0.129 | **1.28** (+28.0%) | 7.41 | **135.15** (-14.1%) |
| **72B Models** | | | | | |
| UI-TARS(Work-Step) | 0.063 | **0.205** | 1.00 | 20.2 | 405.9 |
| UI-TARS(Work-Seq) | 0.063 | 0.180 | **1.25** (+25.0%) | 14.7 | **360.3** (-11.2%) |

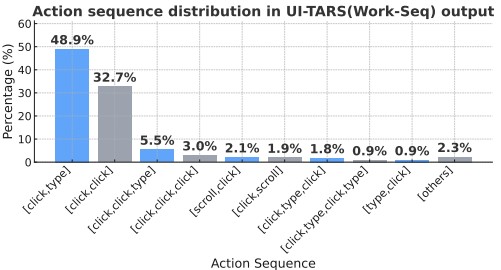

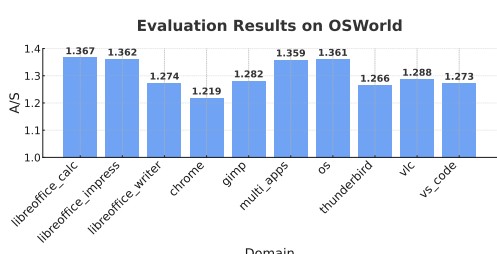

Figure 3: Action-sequence distribution in model output

Figure 4: Evaluation Results on OSWorld

type] for completing two consecutive fields in a form. Overall, this distribution suggests that OS-CATALYST enables the model to generate multi-action sequences in a way that reflects common interaction patterns observed in real GUI tasks. We provide several examples of action sequences in Appendix C.1.

## 5.2 HOW DOES OS-CATALYST PERFORM ON TASK SUCCESS RATE?

On the Seen set, UI-TARS(Work-Seq) achieves success rates that are close to those of UI-TARS(Work-Step), with 0.083 vs. 0.095 for the 7B models and 0.060 vs. 0.079 for the 72B models. This shows that UI-TARS(Work-Seq) method does not compromise the ability to complete tasks. At the same time, UI-TARS(Work-Seq) consistently yields the highest partial success rates, reaching 0.277 (7B) and 0.294 (72B). Compared with the base UI-TARS models without additional training, both fine-tuned variants achieve higher SR and PSR, suggesting that training improves the model's understanding of the WorkArena environment. On the Unseen set, UI-TARS(Work-Seq) achieves comparable success rates to the baselines, with 0.063 SR for both 7B and 72B models. It also sustains strong partial success rates (0.129 and 0.180), showing that OS-CATALYST method generalizes to new task types.

Overall, these results show that OS-CATALYST improves efficiency without reducing task success, and its ability to predict action sequences transfers to tasks outside the training set.

## 5.3 HOW DOES OS-CATALYST PERFORM ON CROSS-DOMAIN SETTINGS?

To assess whether our method generalizes beyond the WorkArena environment, we further evaluate OS-CATALYST on OSWorld. Figure 4 reports the average actions per step (A/S) across different task domains. We observe that OS-CATALYST consistently maintains non-trivial action-sequence prediction ability across unseen environments, achieving A/S values ranging from 1.219 (Chrome) to 1.367 (LibreOffice_Calc). We observe that task domains with more frequent page transitions tend to yield lower A/S values (e.g., chrome, thunderbird), as such interaction patterns limit

opportunities for executing multiple actions within a single interface state. This observation is consistent with human annotation statistics reported in Table 3 of OSWorld-Human (Abhyankar et al., 2025), where grouped steps in several LibreOffice-related tasks show substantially greater reduction compared to single-step interactions, suggesting higher inherent compression potential in such office-style environments.

These results suggest that OS-CATALYST can transfer its adaptive compression capability to heterogeneous GUI environments without retraining, showing potential robustness to domain shifts.

## 6 CONCLUSION

We propose OS-CATALYST, a method that improves the efficiency of computer-using agents through adaptive action compression. By allowing models to predict multiple consecutive actions from a single observation, OS-CATALYST reduces redundant model–environment interactions and shortens overall task duration. To enable this capability, we construct two datasets in the WorkArena environment, supporting both step-level interaction and compressed action sequences. Experiments show that OS-CATALYST achieves up to 50% reduction in task execution time while maintaining comparable task success rates, highlighting the potential of sequence-level action prediction as a new paradigm for GUI agents. Looking ahead, we hope this approach can generalize to broader application scenarios, further advancing the development of efficient and practical GUI agents.

## REPRODUCIBILITY STATEMENT

We provide training and evaluation scripts, together with representative data samples in Work-Seq, in the supplementary materials. Detailed training settings are provided in Appendix B.3. Due to file size constraints, model checkpoints and the complete datasets will be made public to the research community in the camera-ready version.

## ETHICS STATEMENT

Computer-using agents operating on live operating systems could potentially pose risks if not properly constrained. For example, uncontrolled GUI agents may execute unintended operations, misconfigure software, or corrupt sensitive data. Such risks are especially concerning in scientific or enterprise workflows where errors could cause irreversible losses.

In this work, however, all experiments are conducted in isolated benchmark environments (*e.g.*, WorkArena) that contain no sensitive or personal data. Our datasets are constructed from synthetic trajectories generated within these environments, and thus do not involve privacy-related concerns.

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

## A LARGE LANGUAGE MODEL USAGE

In this submission, we leverage LLMs to support and refine the writing process, including grammar and typo correction, and the identification of related work.

## B EXPERIMENTAL DETAILS

### B.1 ENVIRONMENT SELECTION

Among the available GUI benchmarks with diverse features, we selected WorkArena for our experiments. This choice was motivated by the fact that WorkArena tasks generally require a relatively higher number of steps too complete. Moreover, the office scenario naturally lends itself to sequential action execution, making it well-suited for observing how models learn to perform multi-step operations. Following WorkArena, the same team introduced WorkArena++, which incorporates complementary tasks along with more fundamental interactions. However, we found WorkArena++ to be excessively challenging—tasks often exceed 100 steps in length, and preliminary tests showed that both GPT-4o and GPT-4o-v achieved near-zero success rates. Consequently, we decided not to adopt WorkArena++ for this study.

| Action | Definition | Parameter |
|---|---|---|
| click | Clicks at given coordinates. | start_box |
| left_double | Double-clicks at given coordinates. | start_box |
| right_single | Right-clicks at given coordinates. | start_box |
| drag | Drags from start to end position. | start_box, end_box |
| hotkey | Presses a keyboard shortcut. | key |
| type | Types specified content. | content |
| scroll | Scrolls in the given direction. | start_box, direction |
| wait | Pauses for 5s. | / |
| finished | Marks the task as complete. | / |
| call_user | Requests user intervention. | / |

Table 4: Action space with definitions and parameters.

### B.2 ACTION SPACE

We follow the action space design of UI-TARS, while adapting it to our model and dataset. In particular, the action space of the model includes click, left_double, right_single, drag, hotkey, type, scroll, wait, finished, and call_user. The definition and parameter are shown in Table 4.

### B.3 FINE-TUNING SETUP.

We apply the training strategy in Section 3.4 to fine-tune the base models. For the UI-TARS-7B-DPO model, we adopt full SFT for 4 epochs using the ms-swift (Zhao et al., 2024) framework, with a learning rate of $1 \times 10^{-4}$. For the UI-TARS-72B-DPO model, we adopt LoRA-based SFT with rank 8 and train for 4 epochs with the learning rate of $1 \times 10^{-5}$, as full SFT is infeasible under our resource constraints. Here we use LLaMA-Factory (Zheng et al., 2024b) framework for lora fine-tuning.

## C CASE STUDY

In this section, we present representative cases to illustrate the behavior of UI-TARS(Work-Seq). We include both success and failure examples to show how the model generates action sequences in practice.

## C.1 SUCCESS CASE EXAMPLES

Figure 5 and Figure 6 are two success case that demonstrate our model's ability to output consecutive actions. In both examples, the model correctly follows the task instructions and current page state to generate coherent sequences of 3–4 actions.

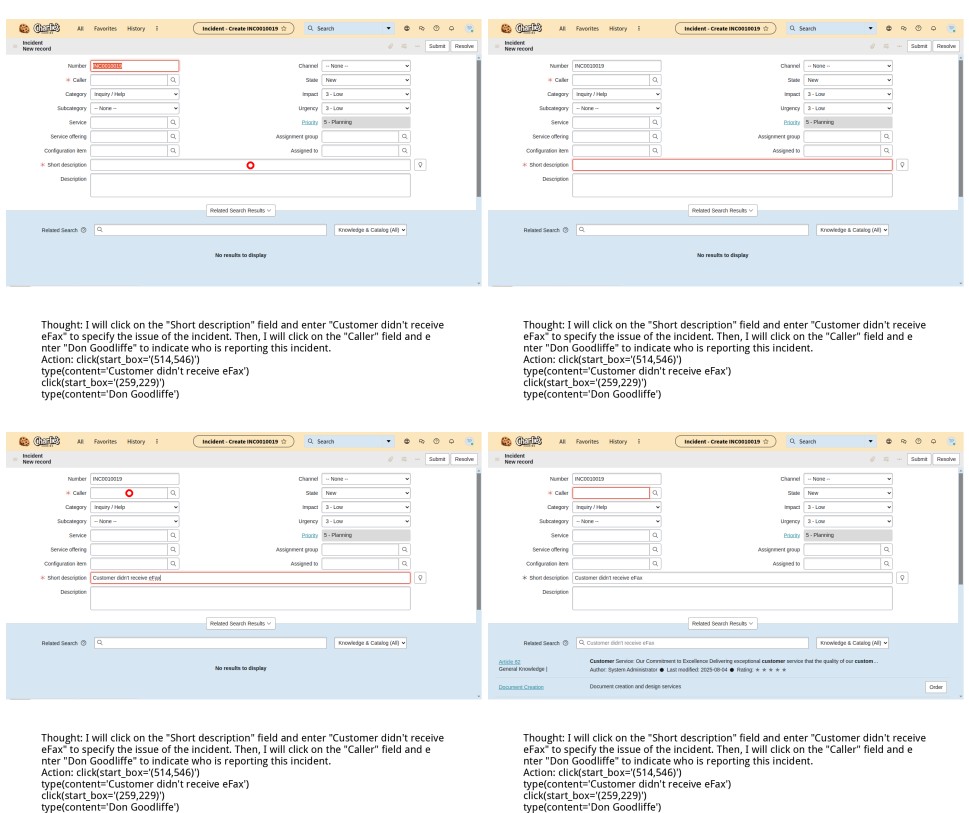

Figure 5: The task is filling up a form. The model output a succession of four actions, filling up two items in a row

## C.2 FAILURE CASE EXAMPLES

We further examine representative failure cases of our model. As shown in Figure 7–9, they can be grouped into three categories: (1) over-compression, where the model outputs an excessively long action sequence beyond what is feasible for the current state; (2) under-compression, where the model fails to merge actions even though multiple steps could safely be combined; (3) incorrect element localization, where the target referenced in the thought is inconsistent with the executed coordinates; These cases illustrate the challenges that remain for robust multi-action planning in GUI environments, and addressing them constitutes an important direction for future work.

While these limitations remain, OS-CATALYST has already led to substantial efficiency improvements over previous methods, reducing overall task time by approximately **50%** and decreasing the average number of interaction steps by **33%**.

## D DATASET DETAILS

In this section, we provide additional details of the datasets used for training in OS-CATALYST. As described in Table 1, our data consists of two subsets: **Work-Step** and **Work-Seq**, both constructed within the WorkArena benchmark environment. Each dataset is designed to support the development of GUI agents from both step-level interaction and action-sequence perspectives.

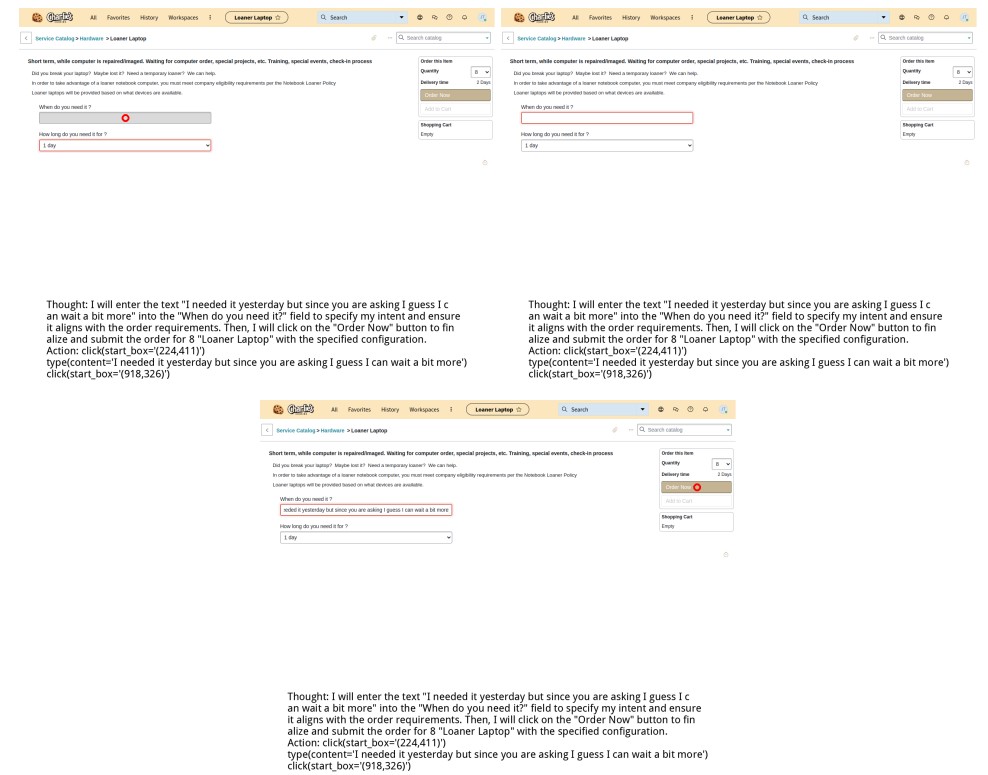

Figure 6: The task is to order a loaner laptop. The model output a succession of three actions, filling up the last item and clicked the submit button

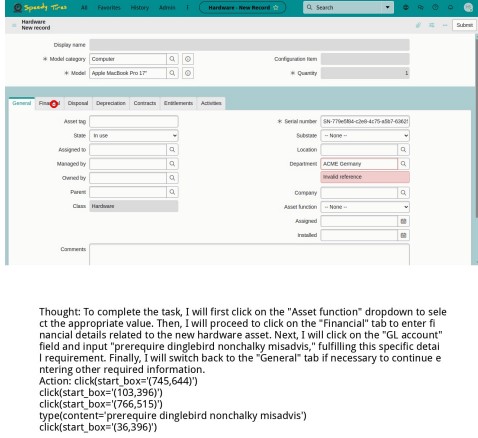

Figure 7: In this form filling task, the model clicks to navigate to the *Financial* subpage. After this transition, it should stop and wait for the environment to return the new page before predicting subsequent actions. However, the model continues to output additional actions prematurely.

**Data Structure.** Each trajectory includes:

- **Screenshots** of the interface at each interaction step.
- **Ground-truth thoughts** that describe the agent's intention and intermediate reasoning.
- **Actions** (e.g., `click`, `type`, `scroll`) along with their coordinates and content.
- **Bounding boxes** specifying the UI elements involved in each action.

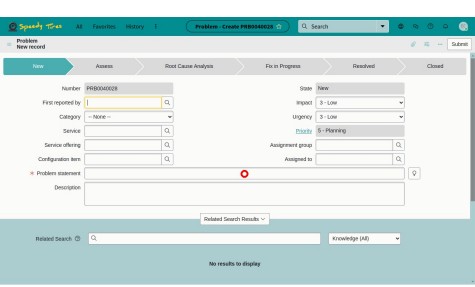 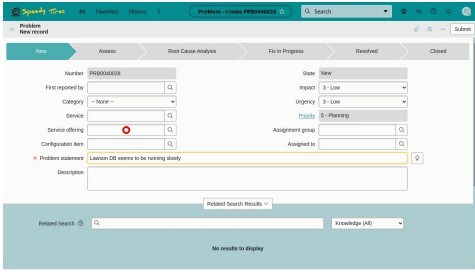

Thought: I will click on the "Problem statement" field and enter the value "Lawson DB seems to be running slowly" to accurately describe the issue as specified in the instruction.
Action: click(start_box='(514,585)')
type(content='Lawson DB seems to be running slowly')

Thought: I will click on the "Configuration item" field and enter the value "lawson_db" to fulfill the requirement for this problem record.
Action: click(start_box='(259,479)')
type(content='lawson_db')

Figure 8: In this form filling example, the model fills in the two form fields across two separate steps, whereas the task can actually be completed in a single step with four consecutive actions.

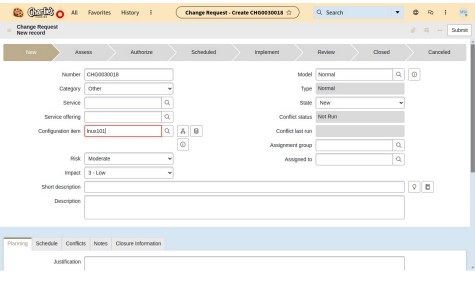

Thought: I will click on the "Closure Information" tab to access and fill out the fields related to closure, such as "Close code" and "Close reason".
Action: click(start_box='(126,43)')

Figure 9: In this example, the correct plan for the model is to click on the *closure information* field. However, the predicted coordinates (marked with a red circle) are far from the correct location.

**Data Format.** The dataset is provided as images and JSON records:

- `*.png`: High-resolution GUI screenshots (1920 × 1080).
- `*.json`: Structured metadata containing thoughts, action definitions, coordinates, and bounding boxes.

**Licensing and Usage.** The dataset will be released under the **MIT License** and can be used for non-commercial academic research, including model training, benchmarking, and GUI agent automation studies. It permits redistribution and modification with proper attribution.

## E PROMPTS

### E.1 MODEL PROMPTS

---

**Original prompt that does not require model to output multiple actions.**

```
You are a GUI agent. You are given a task and your action
history, with screenshots. You need to perform the next action
to complete the task.

## Output Format
```
Thought: ...
Action: ...
```

---

```
```

## Action Space
{action_space}

## Note
- Use {language} in `Thought` part.
- Summarize your next action (with its target element) in one
sentence in `Thought` part.

## User Instruction
{instruction}
```

**Updated prompt that requires model to output multiple actions.**

```
You are a GUI agent. You are given a task and your action
history, with screenshots. You need to perform the next action(s)
to complete the task.

If multiple actions can be performed independently--meaning one
action does not interfere with another in terms of position or
elements--you should output them together in a single `Action`
block, separated by two newlines (`\n\n`).

## Output Format
```
Thought: ...
Action: ...
```

## Action Space
{action_space}

## Note
- Use {language} in `Thought` part.
- Summarize all upcoming actions (with their target elements) in
`Thought` part.
- In the `Action` section, include one or more actions, each on
its own line, separated by two newlines.
- Only include multiple actions if they are **logically and
spatially independent**.

## User Instruction
{instruction}
```

### E.2 DATA CURATION PROMPTS

**Prompt for generating thought.**

```
You are a GUI agent that specializes in reverse-engineering the
intent behind GUI actions.

You will be given a step from an interaction trajectory. Each
step includes:
- the global instruction to complete,
- the previous actions taken,
```

```
- the current action to analyze. (If the current action involves
a coordinate, the coordinates are normalized values: absolute
coordinates divided by the original image width or height, then
multiplied by 1000),
- the UI screenshot with the red bounding box indicating the
position of the action to help you identify the element involved
in the action,
- the UI screenshot after the action is executed,

Your job is to identify the element in the action and infer the
*thought* (i.e., a small plan or rationale) behind the current
action, and then output it in the following format:

Thought: {{<thought>}}

The thought should be a small plan and summarize this action in
future tense (with its target element).
The thought must be consistent with the global instruction and
current action.
The thought should be a plan in a single sentence in
first-person perspective, and it should not include any code or
action.
If the current action is none, and the relevant element is
already set to the correct default that satisfies the instruction
, the thought should state that the default option already meets
the instruction and no further action is needed.

--- INPUT ---

Instruction: {instruction}

Previous actions: {previous_actions if previous_actions else
"None"}

Current action: {current_action}

Current screenshot:
```

**Prompt for judging whether two action can be done in one step**

```
You are given two cropped images of GUI elements. Each image
corresponds to the same position in two consecutive screenshots
from a GUI task execution.

Your task is to determine whether the two images represent the
same GUI element -- that is, the same underlying component such
as a button, icon, text label, or menu item -- even if there are
slight visual differences caused by rendering, state changes
(e.g. hover or focus), or animations.

Minor differences in appearance should not affect your decision,
as long as the core identity of the element remains the same.

Write your reasoning step by step. Then give your final answer
as "yes" or "no" on the last line. ("yes" means both images show
the same GUI element.)
The first element:
```

```
{image1}
The second element:
{image2}
```

**Prompt for merging thoughts of multiple actions**

In the original GUI task setup, the model performs step-by-step
inference: it generates a thought and action, receives an
updated screenshot, and then proceeds with the next thought and
action. The following is a sequence of several consecutive
thought-action steps from that setting and corresponding
screenshots.

Now, we want the model to output all actions in a single step.
Your task is to merge the multiple thoughts into one coherent
and concise thought, as if the model planned the entire sequence
of actions without receiving any updated screenshots in between.

While doing this, remove any reasoning or statements that only
exist due to intermediate screenshots. The final thought should
reflect a continuous reasoning process that naturally leads to
the full sequence of actions without any interruptions.

## Output Format
You should output the merged thought directly in your response,
without any additional text or formatting. The output should be
a single string that combines all individual thoughts into one
coherent and unified thought.

## Previous Thoughts

