# OpenReview forum: "OS-Catalyst: Advancing Computer-Using Agents Efficiency through Adaptive Action Compression"
_ICLR.cc/2026/Conference — ICLR 2026 Conference Withdrawn Submission_

### Official Review · Reviewer_WKML · 2025-10-29

**Soundness:** 2
**Presentation:** 2
**Contribution:** 2
**Rating:** 2
**Confidence:** 4

**Summary:**

This paper propose a data synthesis method to train GUI agents (UI-Tars) to be able to generate more than one action per step to improve efficiency. Specifically, the proposed method recursively merges two consecutive actions when their corresponding screenshots are highly similar and the interacted elements or buttons are independent. This process is achieved using tools such as SSIM, and prompting VLMs such as GPT-4o to check whether the actions can be merged. The authors then constructed such training data on WorkArena, and performed SFT training on UI-Tars on such data. Experiment results on WorkArena shows that this method improves efficiency but yields slightly inferior performance.

**Strengths:**

- The proposed method to construct the "compressed trajectory" is sound. I believe using tools such as SSIM and prompting GPT-4o, actions that are "mergable" can be correctly identified/included in the training data to test the authors hypothesis

- Experimental results on WorkArena shows improved efficiency of the trained models on both seen and unseen subset of the benchmark.

**Weaknesses:**

1. I believe many claims/motivations of this paper is highly specific to the WorkArena benchmark and the UI-Tars models, and may not hold for other models/benchmarks. As an example, L192-193 and L164-177 claims that "models had no awareness of producing action sequences and would not attempt multi-step actions...". This is not true, because models such as DeepSeek-R1/GPT-4o/etc under the default prompt template for OSWorld [1] already produces more than one action per step (see Figure 1 of [2] as an example). Although it is true that many recent computer-use fine-tuned models (e.g., UI-Tars) are trained to output one action per step, it is not generally true for all LLMs/VLMs.

2. The authors constructed by themselves an in-distribution test and out-of-distribution test set with only 84 and 16 tasks respectively. In Table 3, the performance gap between runs are small, yet no standard deviation/statistical significance is provided. In Table 4, all results are only evaluated on 16 tasks (from only 4 task types), which I believe is too small of a test pool to yield any conclusion other than training/testing noise. Overall, I believe the experimental setup for Tables 3 and 4 is poor, making it difficult to draw meaningful insights.

3. Overall, the entire experiments section only uses on benchmark (WorkArena) with testing/training one type of model (UI-Tars). This severely limits the generalizability of the findings: for example, it is unclear whether the conclusions are specific to this model/benchmark setup or would hold for other computer-use models and benchmarks.

---

References:

[1] Xie, Tianbao, et al. "Osworld: Benchmarking multimodal agents for open-ended tasks in real computer environments." Advances in Neural Information Processing Systems 37 (2024): 52040-52094.

[2] Yu, Xiao, et al. "Dyna-Think: Synergizing Reasoning, Acting, and World Model Simulation in AI Agents." arXiv preprint arXiv:2506.00320 (2025).

**Questions:**

- Why is the "step time" higher for "Work-step" models compared to "Work-seq" model? Intuitively, since "Work-step" only needs to generate one action per step, the generated sequence length of this model should be shorter than that of the "Work-seq" model?

---

> ### Author Response · Authors · 2025-11-21
> **Response to Reviewer WKML**
>
> Thank you for your valuable questions and feedback! We appreciate that the reviewer found our compressed trajectory construction method technically sound. We are also encouraged that the improved efficiency on both seen and unseen subsets of the benchmark was positively noted.
>
> Here are the answers to your concerns.
>
> > C1: Comparison with Dyna-think.
>
> A1: We sincerely thank the reviewer for bringing up this relevant work. Their approach demonstrates effective multi-action prediction within a single reasoning step and offers valuable insights.
> However, the referenced approach operates under accessibility-tree mode, where UI elements are represented as structured text. This provides explicit semantic and hierarchical information about the interface. In contrast, our framework is designed for screenshot-based interaction, where the agent must infer UI structure through visual perception alone. This makes multi-action prediction significantly more challenging in our setting, as it requires spatial reasoning, visual grounding, and element correspondence without access to symbolic structure. We list the comparisons as follows:
> | **Aspect** | **OS-Catalyst** | **Dyna-Think** |
> |------------|------------------|----------------|
> | **Observation Modality** | Screenshot-based visual observation | Accessibility-tree–based symbolic UI representation |
> | **Optimization Focus** | Action compression| Reasoning refinement|
> | **Application Scenario** | GUI-based CUA tasks requiring visual grounding | Tasks with structured UI representations suitable for symbolic planning |
>
> Under the constraint of only using screenshot-based visual observations, existing vision-based models are typically limited to performing multiple actions on a single UI element (e.g., clicking and then typing into the same input box). In contrast, our framework enables the agent to predict multi-action sequences that span multiple distinct UI elements within the same interface, based solely on visual context as shown in Figure 4.
>
> We acknowledge that the referenced work provides valuable insights, particularly in reasoning refinement and efficiency optimization. We will cite and include a comparative analysis in the revision to better position our approach relative to others. We sincerely hope that the above clarification helps illustrate that, under the setting of pure visual observation, our framework contributes a meaningful advancement by enabling cross-element multi-action prediction within GUI-based environments which is generally difficult for current vision-based agents to achieve.
>
> > C2: Results on other benchmarks.
>
> A2: Because of time and computational constraints during the rebuttal period, we were not able to complete a full data collection, model retraining, and comprehensive evaluation pipeline. Nevertheless, we conducted a preliminary OOD evaluation by selecting three task domains(117 tasks in total) from the LibreOffice suite in OSWorld and tested our UI-TARS-7B (Work-Seq) model. The model’s efficiency (#A/S) across the domains is shown below:
>
> | **Domain**            | **#A/S** |
> |----------------------|----------|
> | libreoffice\_calc   | 1.367    |
> | libreoffice\_impress| 1.362    |
> | libreoffice\_writer | 1.274    |
>
> Although the model was not fine-tuned on these domains, it demonstrates consistent action-sequence prediction performance in OOD settings. More detailed results and case studies will be included in the revision.
>
> > C3: Why is the "step time" higher for "Work-step" models compared to "Work-seq" model?
>
> A3: We thank the reviewer for this thoughtful question. This difference is mainly due to two reasons:
>
> 1. Context length difference. During inference, we provide up to 5 historical screenshots and messages as context. Since Work-seq compresses multiple actions into fewer steps, a larger proportion of its queries contain fewer than 5 history screenshots. As a result, the average context length is shorter for Work-seq, which reduces its inference time.
> 2. Comparable output text length.
> As illustrated in Figure 2 (new thought generation in action compression), we prompt the model to generate reasoning that is coherent and concise, eliminating unnecessary reasoning.  Therefore, even for multi-action reasoning, the thought sequence for training is compact rather than proportionally longer.
> Moreover, we measured the output length during evaluation and found that it remains comparable across settings. For example, using the 72B model, the average number of characters generated per query is 239.55 for Work-seq versus 256.34 for Work-step. This confirms that multi-action prediction does not lead to longer generation in our method.

---

> ### Comment · Reviewer_WKML · 2025-11-24
>
> Thanks for the response. Overall I believe the responses does not address my three concerns raised in the weaknesses section.
>
> ---
>
> Regarding concern 1, I note that I said "I believe many claims/motivations of this paper is highly specific to the WorkArena benchmark and the UI-Tars models, and may not hold for other models/benchmarks." I used an example from Dyna-Think to illustrate the point, but I did not say anything about "Comparison with Dyna-think". *My point is this work's motivation, analysis, and results are focused on one setting (WorkArena benchmark and the UI-Tars models). It is questionable whether the claims made in this work hold generally for computer-use agents/environments.*
>
>
> Regarding concern 2, this is not addressed at all. "Results on other benchmarks" corresponds to my 3rd point in the weaknesses section. Additionally, why is only the LibreOffice domain from OSWorld chosen instead of other domains such as OS, thunderbird, chrome, or VSCode? By "we conducted a preliminary OOD evaluation", I assume you mean directly evaluating your trained models on OSWorld. Since just running evaluation should be straightforward, is there any reason for only evaluating on the LibreOffice domain?
>
>
> Regarding concern 3, this is from the questions section, not from the weakness section. I appreciate your response, but I believe your notation of "C1", "C2", and "C3" is highly confusing as they do not correspond to the numbering I provided in my original review.

---

> > ### Author Response · Authors · 2025-11-30
> > **Response to Reviewer WKML**
> >
> > Thank you for your detailed review and constructive suggestions. We also appreciate your reminder regarding the numbering in our previous response. As the question section has already been addressed extensively, we now provide clarifications corresponding to each point in the Weaknesses section as originally stated.
> >
> > > Weakness 1 — Generality of motivation and claims
> >
> > Our motivation and claims are formulated for general vision-based GUI environments and agents. We selected WorkArena as a representative GUI benchmark and UI-TARS as a widely-used model in this setting to conduct our initial exploration. These choices provide a realistic and practically relevant testbed, rather than implying any restriction of our method to this particular setup. To further support the generality of our findings, we additionally evaluate on OSWorld (see response to Weakness 3).
> >
> > > Weakness 2 — Scale of evaluation tasks
> >
> > The in-distribution (84) and out-of-distribution (16) test splits may seem limited in absolute terms. However, tasks in existing GUI benchmarks (such as AndroidWorld and OSWorld) are typically in this scale (a few hundred total annotated tasks per benchmark). Based on prior literature, we consider this sufficient for evaluation and analysis in GUI agent settings. Additionally, we provide further out-of-distribution evaluation across OSWorld domains (see Weakness 3 response), which strengthens the generality of our findings.
> >
> > > Weakness 3 — Evaluation on other benchmarks
> >
> > We reported the LibreOffice results previously because OSWorld-Human[1] statistics indicate higher step compression potential in office-style GUI tasks, making this domain a natural starting point to demonstrate model capability. In response to the generality concern, we now extend our evaluation to additional OSWorld domains, without any fine-tuning:
> >
> > | **Domain**            | **#A/S** |
> > |----------------------|----------|
> > | libreoffice\_calc   | 1.367    |
> > | libreoffice\_impress| 1.362    |
> > | libreoffice\_writer | 1.274    |
> > | chrome | 1.219    |
> > | gimp | 1.282    |
> > | multi_apps | 1.359    |
> > | os | 1.361    |
> > | thunderbird | 1.266    |
> > | vlc | 1.288    |
> > | vs\_code | 1.273    |
> >
> > The consistent results across domains demonstrate that OS-Catalyst is not confined to WorkArena, and exhibits potential generalizability within vision-based GUI environments.
> >
> > Thank you again for your insightful feedback. We also revised the manuscript to incorporate the new cross-domain evaluation results.
> >
> > [1] Abhyankar, et al. "OSWorld-Human: Benchmarking the Efficiency of Computer-Use Agents." arXiv preprint arXiv:2506.16042, 2025.

---

### Official Review · Reviewer_q5vT · 2025-10-29

**Soundness:** 3
**Presentation:** 3
**Contribution:** 2
**Rating:** 6
**Confidence:** 3

**Summary:**

In this paper, the authors focus on the problem of developing effective agents that can navigate computer user interfaces. However, one issue with this domain is the action space is usually very large and complex. To address this problem, the authors propose OS-Catalyst, an approach with compresses sequences of actions via grouping them by similarity and region checks. They also construct a dataset for evaluation. They show promising results and notable agent speedup.

**Strengths:**

1. The approach is simple and easy to understand.

2. The paper is well-written and well presented.

3. Quantitative performance is promising

4. Evaluation setup is thorough and broad based.

**Weaknesses:**

1. Technical novelty is incremental. This is largely about a simple approach to action compression within a pre existing framework.

2. While performance is impressive, the speedup is somewhat modest (about 2x).

**Questions:**

1. Can you please elaborate on the technical novelty of the approach?

2. Are there any tweaks that can be made to the approach for more significant speedups (> 5x) without dramatically sacrificing performance?

---

> ### Author Response · Authors · 2025-11-21
> **Response to Reviewer q5vT**
>
> We highly appreciate your valuable suggestions. We are sincerely grateful that the reviewer finds our approach intuitive and easy to follow, acknowledges the clarity and quality of our presentation, and views the results and evaluation setup as promising and comprehensive. Your positive feedback is very encouraging to us.
>
> Here are the answers to your concerns.
>
> > C1: This is largely about a simple approach to action compression within a pre existing framework.
>
> A1: While compressing seems straightforward in hindsight, we would like to clarify that the core contribution lies not in a single compression heuristic, but in establishing a systematic paradigm for adaptive multi-action prediction in GUI agents. This includes (i) a dedicated data construction and compression pipeline, (ii) the empirical demonstration that action-sequence prediction is both feasible and highly effective in realistic GUI environments. Experimental results on WorkArena show that our method achieves up to 50% reduction in task execution time without sacrificing success rate, indicating that our approach is not merely an optimization, but a paradigm-level improvement toward efficient GUI agents.
>
> > C2: While performance is impressive, the speedup is somewhat modest (about 2x).
>
> A2: The overall task execution time can be roughly estimated as the product of the number of steps and the time per step:
>
> $$
> \text{Total Time} \approx (\text{Steps}) \times (\text{Step Time})
> $$
>
> - **Steps**: The number of steps is limited by **page transitions**, which we found can only be reduced by up to one-third in WorkArena environment due to the inherent characteristics of GUI tasks.
>
> - **Time Per step**: The time per step is constrained by infra/system limitations. While our method focuses on improving the number of steps, it does not address infra/system delays, which remain a limiting factor.
>
> Currently, we achieve approximately a 2x speedup through step reduction, which we believe already represents a substantial improvement. Further acceleration (e.g., beyond 5x) would require system-level and hardware optimizations, as additional gains are constrained by interaction latency, environment feedback, and inference delay—factors outside the scope of algorithmic step compression. Our future work may explore potential optimizations in infrastructure and system design to achieve more significant speedups.

---

### Official Review · Reviewer_EPo9 · 2025-10-31

**Soundness:** 3
**Presentation:** 3
**Contribution:** 2
**Rating:** 4
**Confidence:** 3

**Summary:**

This paper proposes OS-Catalyst, a system designed for reducing the execution latency of computer-use agents. The observation is that multiple actions can often be conducted together (by merging them) based on a single observation. The authors then described their data construction using the WorkArena benchmark and how they use it to fine tune the UI-TARS model. By changing sequential actions (executed step by step) to a grouped action execution, OS-Catalyst achieves up to 50% reduction in total task execution time on the WOrkArena benchmark.

**Strengths:**

- The paper performs a comprehensive methodology from constructing trajectory training dataset manually to fine tune the UI-TARS model to evaluate it.
- The result is very encouraging, achieving up to 50% latency improvement.
- Presentation is overall clear

**Weaknesses:**

- Only evaluated on WorkArena. It's unclear how well the proposed method would work on other types of CUA tasks like OS commands, CLIs, etc.
- Action merging is based on heuristics and depends on thresholds, and there's no sensitivity test for these thresholds.
- The paper talks about over-compression, under-compression, etc, but does not propose concrete solutions to address them
- The proposal of grouping CUA actions and the construction of action-group trajectories are not new. OSWorld-Human (https://arxiv.org/abs/2506.16042) performs action grouping and adds manual trajectories to the OSWorld benchmark.

**Questions:**

Can you explain how the thresholds are set?

---

> ### Author Response · Authors · 2025-11-21
> **Response to Reviewer EPo9**
>
> We sincerely appreciate your inspiring suggestions. We are grateful that the reviewer acknowledges our effort in exploring latency reduction, the comprehensiveness of our methodology as well as the encouraging improvement.
>
> Here are the answers to your concerns.
>
> > C1: Evaluation results on other benchmarks.
>
> A1: We thank the reviewer for raising this insightful point.
> Because of time and computational constraints during the rebuttal period, we were unable to perform the full data collection, training, and evaluation pipeline. However, we directly conducted an OOD evaluation by selecting the following task domains from the LibreOffice suite in OSWorld[1] and tested our UI-TARS-7B[2] (Work-Seq) model. The model’s efficiency (#actions per step) across the domains is reported below:
>
> | **Domain**            | **#A/S** |
> |----------------------|----------|
> | libreoffice\_calc   | 1.367    |
> | libreoffice\_impress| 1.362    |
> | libreoffice\_writer | 1.274    |
>
> Despite not being fine-tuned on these domains, the model maintains effective action sequence prediction performance in OOD settings. We will include detailed results and case studies in the expanded experimental analysis section in our revision.
>
> Our method is specifically designed for GUI-based CUA tasks, where:
> 1. The agent operates on visual observations and executes grounded dependent actions. This makes compression more challenging, as multi-step reasoning must be inferred within a single observation.
> 2. In contrast, CLI or OS command tasks interact through symbolic interfaces, where the command already encodes the complete intent in a single structured input.
> 3. CLI-based tasks are inherently more efficient, as they avoid the need for visual parsing and generating step-level actions. Thus, we hold the view that action compression is unnecessary in this setting and such tasks can be seamlessly integrated with our framework.
>
> > C2: Settings of thresholds.
>
> A2: The SSIM threshold is used for preliminary step to check for large page changes between consecutive steps, but it is not critical to the overall pipeline. The actual decision to merge actions is made during the region match step, where we use GPT-4o to verify if the second action's target element remains consistent. We manually sampled and verified GPT-4o's ability to make these judgments, confirming that it can reliably perform this task.
>
> > C3: Solution of address over/under compression.
>
> A3: We would like to clarify that effective compression is currently difficult to achieve for most existing models. Compared to previous approaches, our method has already made significant improvements in efficiency(total time -50%, #Actions/Steps -33%) by reducing redundant steps and optimizing action sequences. While there remains room for further refinement, we view OS-Catalyst as an early exploratory attempt in this direction, and leave the development of more advanced compression strategies as future work.
>
> > C4: Comparison with action-grouping.
>
> A4: Thank you for your suggestion to compare our method with OSWorld-Human. This comparison is very insightful, and we appreciate the opportunity to clarify the similarities and differences between the two works.
>
> | **Aspect** | **OSWorld-Human** | **OS-Catalyst** |
> |------------|-------------------|-----------------|
> | **Focus on GUI agent efficiency optimization** | ✓ | ✓ |
> | **Data annotation** | Manual annotations | Automated pipeline |
> | **Role** | Benchmark | Method |
> | **Focus** | Evaluation metrics | Method pipeline, dataset, and results |
> | **Improvement proposed** | ✗ (No improvement methods proposed) | ✓ (Training and performance validation) |
>
> As seen in the table:
> 1. Both methods focus on improving GUI agent efficiency.
> 2. OSWorld-Human uses manual annotations for action grouping, whereas OS-Catalyst relies on an automated pipeline.
> 3. OSWorld-Human serves as a benchmark, emphasizing evaluation metrics, whereas OS-Catalyst is a method, with a focus on the training process, automated pipeline, and performance validation.
> 4. While OSWorld-Human does not propose improvements, we introduce a full training and validation process to demonstrate the effectiveness of our method.
>
> We hope this comparison clarifies the distinct focus and contributions of each approach.
>
> [1] Xie, Tianbao, et al. "Osworld: Benchmarking multimodal agents for open-ended tasks in real computer environments." Advances in Neural Information Processing Systems 37 (2024): 52040-52094.
>
> [2] Yujia Qin, et al. Ui-tars: Pioneering automated gui interaction with native agents. arXiv preprint arXiv:2501.12326, 2025b.

---

> > ### Comment · Reviewer_EPo9 · 2025-11-27
> >
> > Thank you for your detailed response. I'm willing to raise your score to a 5 or 6, accordingly. (although I haven't found how to update my overall raiting score). I hope AC can take into consideration of my comment here.

---

> > > ### Author Response · Authors · 2025-11-30
> > > **Response to Reviewer EPo9**
> > >
> > > Thank you very much for your positive feedback and for considering increasing the score of our paper. We sincerely appreciate your constructive suggestions.
> > >
> > > We have updated the manuscript and uploaded the revised PDF, incorporating the additional experiments conducted during the rebuttal period and updating the Related Work section. Thank you again for your valuable support and encouragement.

---

### Official Review · Reviewer_F7dh · 2025-10-31

**Soundness:** 2
**Presentation:** 2
**Contribution:** 2
**Rating:** 2
**Confidence:** 3

**Summary:**

The authors address the latency problem in step-by-step computer-using agents. The authors propose OS-CATALYST, a framework based on adaptive action compression, which predicts and executes sequences of actions jointly rather than step-by-step. This approach aims to reduce the number of model queries while maintaining task success.
They also introduce a new dataset tailored for this task. Experimental results show that OS-CATALYST achieves approximately 50% faster execution with no loss in success rate compared to baseline agents.

**Strengths:**

- The authors are transparent about their use of LLMs, including a clear statement clarifying the extent of model usage.
- They share a supplementary ZIP file containing partial dataset information, code fragments, and training scripts, which supports reproducibility to some degree.

**Weaknesses:**

- Although supplementary materials are provided, there is no README or documentation to explain their structure or usage, making it difficult to reproduce or extend the experiments.
- The dataset, which is presented as one of the main contributions, is not clearly or comprehensively described. It deserves a dedicated section detailing its composition, sources, scale, and intended use cases.
- The main paper **exceeds the 9-page limit**, which could be grounds for a desk rejection under ICLR formatting rules.

**Questions:**

- One of the key contributions is the dataset. Could the authors provide a clearer overview and discuss its size, structure, and licensing terms in more details? It may be helpful to dedicate a separate section for this.

---

> ### Author Response · Authors · 2025-11-21
> **Response to Reviewer F7dh**
>
> Thank you for your valuable feedback and questions! We appreciate that the reviewer recognized our efforts in mitigating latency via adaptive action compression and acknowledged our transparency in LLM usage.
>
> We will make the following revisions to our paper:
> 1. We will add details on the dataset structure and licensing terms in the main paper to ensure full clarity.
> 2. We will provide a README file in the supplementary materials, which will outline the dataset’s composition and usage.
>
> Here are the answers to your concerns.
>
> > C1: The main paper exceeds the 9-page limit, which could be grounds for a desk rejection under ICLR formatting rules.
>
> A1: The reviewer appears to consider that our submission exceeds the page limit; however, we would like to clarify that the Ethics Statement section appears on the 10th page in our submission. According to the ICLR author guidelines, the Ethics Statement does not count towards the 9-page limit. Therefore, our submission complies with the page limit requirement. We really appreciate your understanding and hope this clarification resolves any concerns regarding the length of the paper.
>
> For reference, here are the relevant guidelines from the ICLR Author Guide:
>     "The optional ethics statement will not count toward the page limit, but should not be more than 1 page."(https://iclr.cc/Conferences/2026/AuthorGuide)
>
> > C2: Reproducibility of the dataset.
>
> A2: As shown in Table 1 in the paper, we provide key information about the dataset, such as the number of trajectories and actions per step. Our dataset consists of two subsets: Work-Step and Work-Seq. Both subsets are designed to support the development of GUI agents. Each trajectory contains:
> - Screenshots at each step,
> - Ground truth thoughts explaining the actions,
> - Actions performed by the model (clicks, typing, etc.) and corresponding coordinates,
> - Bounding boxes identifying relevant UI elements.
>
> Our dataset, which includes images and corresponding JSON-format data, is for non-commercial academic use, including training, benchmarking, and research on GUI agent training automation and action sequence optimization. It is licensed under the MIT License, allowing free use, modification, and distribution with proper attribution.

---

> > ### Comment · Reviewer_F7dh · 2025-11-27
> > **Reply**
> >
> > Thank you for your detailed responses to my questions. I appreciate your efforts in clarifying the points that were causing confusion for me. I now have a much better understanding of most of the aspects that were unclear.
> >
> > To be able to reevaluate the paper contribution, I would like to see the revised version of the paper. Would it be possible to include the revised version of the paper? It would help continue the discussion.

---

> ### Author Response · Authors · 2025-11-30
> **Response to Reviewer F7dh**
>
> Thank you very much for your response and for taking the time to review our work.
> We have now uploaded the revised version of the paper, including a newly added section of dataset details in the Appendix to further clarify the information about the dataset.
> We sincerely appreciate your constructive comments and support.

---

### Author Response · Authors · 2025-11-24
**We sincerely thank all reviewers for their thoughtful and constructive feedback.**

We sincerely thank all reviewers for their thoughtful and constructive feedback. We are encouraged by the positive recognition of our work, including the effectiveness of our adaptive action compression strategy in reducing execution latency (approximately 50% improvement) `EPo9`, the clarity and soundness of our methodology `q5vT` and presentation `EPo9` `q5vT`, as well as the promising quantitative results demonstrated on both seen and unseen benchmark tasks `WKML`. We also appreciate the reviewers’ acknowledgement of our transparency in LLM usage and the degree of reproducibility `F7dh` enabled by sharing dataset resources and implementation details. These comments are highly motivating and reinforce the value of our contribution.

In our individual responses, we have responded to each concern raised by the reviewers and provided detailed clarifications where relevant. We hope these explanations will help resolve the questions raised. Additionally, we will revise the manuscript to improve technical rigor and presentation clarity. The planned revisions are summarized as follows:

1. Dataset details and reproducibility. We will add information on dataset structure and licensing terms in the main paper and include a supplemental README to outline dataset composition and usage guidelines. `E7dh`
2. Expanded experimental analysis. We will incorporate additional OOD evaluation results (e.g., LibreOffice domains from OSWorld) and include case studies in the revised experimental section. `EPo9` `WKML`
3. Comparison with existing approaches. A more explicit comparative discussion with OSWorld-Human and Dyna-Think will be added, clarifying the differences in observation modality, optimization objectives, and application scenarios. `EPo9` `WKML`
4. Statement of future work. We will update the manuscript for clarity, consistency, and precision, particularly refining the description of limitations and future directions related to over/under compression. `EPo9`

We welcome any further discussion and feedback from the reviewers to help us continue improving this work.

---

### Note · Authors · 2026-01-06

I have read and agree with the venue's withdrawal policy on behalf of myself and my co-authors.